# Disparate estimates of intrinsic productivity for Antarctic krill across small spatial scales under a rapidly changing ocean

Mauricio Mardones[1,2,3]*, Erica Jarvis Mason[4], Francisco Santa Cruz[1,2,5], George Watters[4], César Cárdenas[2,5]*

**1** Doctorate Program Antarctic and Subantarctic Science, Universidad de Magallanes, Punta Arenas, Chile, **2** Millennium Institute Biodiversity of Antarctic and Subantarctic Ecosystems, Santiago, Chile, **3** Centro Ideal FONDAP, Universidad Austral de Chile, Valdivia, Chile, **4** Antarctic Ecosystem Research Division, Southwest Fisheries Science Center, NOAA, La Jolla, California, United States of America, **5** Departamento Científico, Instituto Antártico Chileno, Punta Arenas, Chile

* mamardon@umag.cl (MM); ccardenas@inach.cl (CC)

## Abstract

Understanding the spatio-temporal dynamics of Antarctic krill (*Euphausia superba*) productivity along the Western Antarctic Peninsula (WAP) requires the application of robust analytical approaches. Both design-based and model-based methodologies have been employed to address this challenge. The Spawning Potential Ratio (SPR) is a indicator that provides valuable insights about population productivity. In this study, we analyzed the spatial and temporal variability of the SPR for krill by modeling Length-Based Spawning Potential Ratio (LBSPR) with 20 years of fishery-dependent length composition data. Results showed spatial and temporal heterogeneity among five fishing strata in the WAP, where SPR in Gerlache Strait stratum was consistently lower than the 20% SPR reference point, compared with Elephant, Bransfield Strait, Southwest and Joinville Island strata. Moreover, we demonstrate the sensitivity of LBSPR to changes in growth parameters, such as $k$ and $L_{inf}$, which are influenced by environmental variables like chlorophyll. Our findings underscore the value of incorporating environmental variability into stock assessment models, such as those based on SPR, to accurately assess krill stock conditions. Given the apparent spatial heterogeneity in intrinsic productivity identified through our SPR estimates, we propose using this approach to establish a management procedure based on a control rule for each stratum. This method adjusts the allocation of catch limits in line with the new management strategy of the Commission for the Conservation of Antarctic Marine Living Resources (CCAMLR). By integrating knowledge about spatial krill dynamics and its intrinsic productivity, advice can be recommended to promote the sustainable management of krill populations in Subarea 48.1.

**Data availability statement:** The original raw data underlying this study are managed by the Commission for the Conservation of Antarctic Marine Living Resources (CCAMLR) and are subject to legal and institutional restrictions under the CCAMLR Rules for Access and Use of Data. As a result, the authors do not have permission to publicly share the original raw datasets. However, the minimal processed and anonymized datasets required to replicate the analyses and results presented in this study are publicly available in a stable repository. These processed data, together with the complete analytical workflow, have been deposited in Zenodo and can be accessed via the following DOI: https://doi.org/10.5281/zenodo.17936869. Also, all source code used to generate the results is publicly available at https://github.com/MauroMardones/LBSPR_Krill and is linked to the Zenodo archive. The processed datasets archived in Zenodo consist of aggregated and atomized length–frequency data derived from the original CCAMLR datasets exclusively for the purpose of reproducibility and do not substitute the original raw data holdings.

**Funding:** CC; FS: INACH "Marine Protected Areas" Program (Grant No. 2409052). CC: ANID/Millennium Science Initiative Program (Grant No. ICN2021_002). MM: Doctorate Scholarship from ANID/Centro IDEAL, FONDAP at the Universidad Austral de Chile. (Grant No. 15150003) MM: Commission for the Conservation of Antarctic Marine Living Resources (CCAMLR) Scholarship Scheme (2023-2024) The funders had no role in study design, data collection and analysis, decision to publish, or preparation of the manuscript.

**Competing interests:** The authors have declared that no competing interests exist.

## Introduction

Antarctic krill (*Euphausia superba*, hereafter krill) is a key species in the Southern Ocean food web [1,2]. It supports many predators and also sustains an important commercial fishery. Its population dynamics are strongly influenced by environmental and oceanographic variability [3,4]. The Western Antarctic Peninsula (WAP) is one of the most climate-sensitive areas in the Southern Ocean and has experienced rapid changes across multiple dimensions [5]. Over the last 40 years, climate driven changes have resulted in warming waters [2,5], declines in seasonal sea ice extent and duration [6,7] and changes in phytoplankton productivity [2,8,9]. WAP is a critical region for krill productivity, serving as a major spawning and recruitment area, and also, as a vital region for breeding and foraging by many cetaceans, pinnipeds and seabirds [10–13]. Given these characteristics, it is expected that environmental changes in this area have influenced the spatial and temporal dynamics of the krill population [14]. Furthermore, these environmental changes operate at different scales, resulting in localized spatio-temporal shifts in the krill population within the WAP [15–17].

Krill is considered largest marine biomass on Earth [4,10,15] and has been the subject of increasing commercial exploitation over the last 50 years [10,18–21]. Since the early 1990s, the fishery has been concentrated between the WAP and the Scotia Sea (FAO statistical subareas 48.1, 48.2 and 48.3) where about 70% of the krill population resides [4,9,10,17,22]. The Commission for the Conservation of Marine Living Resources (CCAMLR) is a decision-making body aims to conserve Antarctic fish stocks, including krill, and associated ecosystems using the best available science. Specific objectives of CCAMLR are to minimize the risks associated with harvest rates that may affect the populations and avoid irreversible impacts on the ecosystem [23–27 Art. II].

The krill fishery management scheme implemented by CCAMLR establishes catch limits in the Convention Area based on a harvest control rule that depends on two biomass reference points [22,23,25,28]. The first rule (*"the depletion rule"*) is based on the lowest level of harvesting allowed, around 20% of pre-harvest biomass, which could be considered a limit reference point. The second rule (*"the escapement rule"*) is the target level of the fishery, which indicates the statistical distribution of the biomass at the end of the 20-year projection under a constant catch that allows the median escapement of 75% at pre-harvested levels of biomass. This scheme resulted in the set of a regional precautionary catch limit within the WAP area, based on a synoptic estimates of biomass from 1991, 2000 and 2019 [26,29].

Many fisheries regulate catch limits using feedback control rules [30–32] centered on Biological Reference Points (BRPs). In contrast, CCAMLR does not apply a feedback control mechanism to krill management [13,25,28,30,33]. With a long-standing Conservation Measure (CM 51−07) [34] now expired due to the lack of consensus, the absence of an adaptive management procedure may compromise the effectiveness of krill regulation and, consequently, the sustainability of the resource, particularly under spatially uneven fishing effort and environmental variability. In fact, in Subarea 48.1, where a significant krill fishing activity occurs, concerns remain that

even at this regional scale, the catch limit may be insufficient to prevent localized impacts on krill and indirect effects on krill predators [3,26]. To address these issues, CCAMLR is attempting to revise its management strategy and conservation measures for the krill fishery. The challenge is to enhance resource management and conservation by adopting a more precautionary and ecosystem-based strategy [4,13,35]. In this new management strategy, the spatial and temporal changes in the krill population and community structure are relevant because management at local or reduced spatial scales, instead of regional, has been proposed [13,26,36]. A significant obstacle to achieving small-scale management is the absence of comprehensive quantitative performance metrics, based on BRPs, that can aid assessment of the degree to which factors influencing spatial heterogeneity have affected krill populations within the WAP [25,26]. This new management strategy should incorporate spatial considerations and account for both krill population dynamics and ecosystem integrity [22,25,26].

Understanding krill population changes is crucial, as these shifts, reflected in movement patterns, ontogeny, and biological traits, directly influence the spatial and temporal productivity and intrinsic productivity of krill populations in the WAP [12,37,38]. The intrinsic productivity refers to the ability of a population to reproduce and sustain itself, which depends on reproductive and somatic conditions [39,40]. A common way to measure intrinsic productivity is through the Spawning Potential Ratio (SPR). The SPR compares the reproductive output of a fished population to that of an unfished population, and depends mainly on growth and fishery selectivity [41–47]. The SPR can indicate the status of a fished stock in areas with variable exploitation rates, because recruitment can be related to overfishing [48,49]. That is, when the reproductive fraction has been severely reduced due to excessive fishing, the success of recruitment and stock replenishment is jeopardised [48,50,51]. The SPR has been widely used due to the advantages related with its easy biological interpretation and also because it uses the length composition of the catches, one of the most abundant and reliable sources of information obtained from fishing activity [41,43,52]. Numerous length-based assessment methods have been used to quantify alterations in intrinsic productivity in harvested populations, specifically the SPR [41,44,47,53]. For example, the Length-Based Spawning Potential Ratio (LBSPR) [44] is a modeling approach to derive SPR and relative fishing pressure estimates from length composition and life history data [43,44,46,54,55,55]. Given that the SPR can be used as an indicator of population status, we can use it to improve our understanding of the degree of spatial heterogeneity of the krill population in the WAP. This may consequently, provide elements for establishing limit and target BRPs and allow for flexible adjustment of catch limits in krill fishery management [49,56]. In addition to fishing, environmental factors play a crucial role in shaping the SPR by affecting essential life history traits of krill [38,57]. Fluctuations in temperature, salinity, and nutrient availability directly affect metabolic rates, growth, and reproductive output [3,58–63,63–65]. In marine environments, elevated temperatures can accelerate metabolism, potentially increasing growth but also demanding greater energy expenditure [8,18,65]. Conversely, nutrient limitation can constrain primary production, reducing food availability and consequently, impacting population growth [8,63].

This study aims to quantify the intrinsic productivity of krill using LBSPR modeling using data collected over the past two decades across five strata currently identified by CCAMLR in Subarea 48.1 [28,66]. Additionally, we assess changes in somatic growth, which are influenced by environmental conditions, and test their impact on the intrinsic productivity of the krill population. The results of this spatially explicit, strata-based assessment of productivity using SPR as a key indicator, offer valuable insights to refine and optimize catch limit allocations in WAP. These adjustments are based on krill BRPs to support the sustainable management of the krill fishery in Subarea 48.1, aligning with CCAMLR's revised fishery management strategy.

## Methodology

### Study area

The study area corresponds to Subarea 48.1 in the WAP, where most krill fishing activity occurs and where CCAMLR has been working to implement a new management strategy at a finer spatial scale. To achieve a more precise spatial definition of

krill population dynamics and analysis, we considered the five strata proposed by CCAMLR: Bransfield Strait, Elephant Island, Gerlache Strait, Joinville Island, and the Southwestern South Shetland Islands (hereafter Southwest) (Fig 1). It is worth noting that although these strata were originally delineated from oceanographic and practical considerations and did not initially incorporate biological or population-specific characteristics, they nonetheless provide a suitable framework for our analysis.

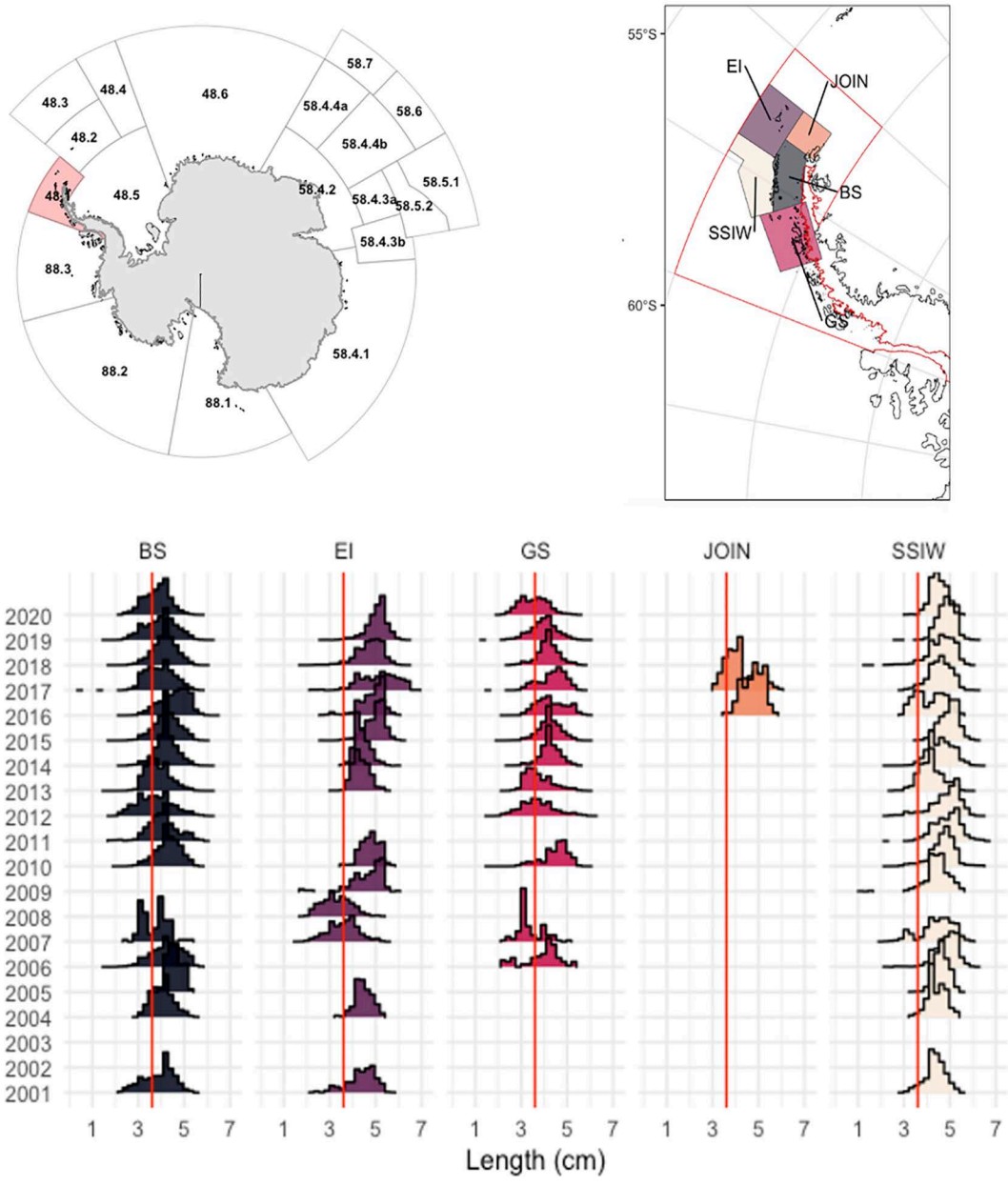

**Fig 1. Spatio-temporal patterns in the length distribution of Antarctic krill (Euphausia superba) across five subareas of the Antarctic Peninsula region.** Top left: CCAMLR statistical subareas around Antarctica, highlighting Subarea 48.1. Top right: Management units in 48.1 subarea considered in this study: Bransfield Strait (BS), Elephant Island (EI), Gerlache Strait (GS), Joinville Island (JOIN), and the Southwestern South Shetland Islands (SSWI).). Bottom: Annual krill length composition (2001–2020) for each management unit. Red line is length recruit (36 mm). Base map data from Natural Earth (public domain).

## Monitoring data (CCAMLR SISO)

We obtained data from the monitoring of the krill fishery, which has been systematically monitored onboard fishing vessels by scientific observers as part of the CCAMLR Scheme of International Scientific Observation (SISO). Krill length compositions (total length, TL) measured from the anterior edge of the eye to the posterior end of the telson, excluding terminal spines) comprising 685,745 individual records were obtained from Subarea 48.1 for the 2000–2020 period, and grouped by month, year and management stratum in 2 mm length bins (Fig 1). The database included information on vessel identification, nationality, georeferenced fishing locations, among other fields. To enhance the representation of size-based population structure for subsequent modeling, descriptive statistics were calculated using the *"tidyverse"* package in R [67,68].

## Environmental drivers of krill growth

To establish environmental drivers of krill growth in the WAP, we focused on the relationships between krill length and three key environmental variables, identified in previous studies as the most influential factors in krill population dynamics [2,8,18,38,69]: 1) sea surface temperature (SST) using ERA5 monthly mean data from 1991 to present; 2) sea ice concentration (SIC) using Nimbus-7, from November 1978 to present and 3) chlorophyll-a (Chla) data from Bio-Geo-Chemical, L4 (monthly and interpolated) Satellite Observations. All data from satellite observations in native resolution were downloaded using E.U. Copernicus Marine Service Information (doi.org/10.48670/moi-00021; doi.org/10.48670/moi-00148) and using native resolution, which represents the finest level of detail captured by the measurement instrument or sampling method. Post-processing techniques, including data cleaning and handling algorithms, manipulation of spatial data, rasterization and projection transformations were facilitated by the *sf* and *ncdf4* packages in R [70,71].

We tested for correlations between krill length compositions from the fishery data and environmental variables using Pearson correlation tests. The Pearson correlation coefficient (*r*) indicates both the strength and direction of the linear association between two continuous variables [70]. Based on the results of collinearity among variables, the linear models described below were constructed. Also, to analyze the factors influencing krill growth, we used a Generalized Linear Mixed Model (GLMM) framework [72]. The response variable in the model is *Length*, which serves as a proxy for krill growth. In fisheries science, length is widely recognized as a reliable proxy for growth because it reflects somatic development and underpins the most influential life-history and stock-assessment frameworks. Foundational models such as the von Bertalanffy Growth Function (VBGF) explicitly depend on observed lengths to estimate key parameters [73–77], while modern length-based approaches further reinforce its central role in describing productivity and biological status. This approach allows us to account for both fixed and random effects, capturing spatial and environmental variability while addressing potential non-independence in the data. The linear model is formulated as follows:

$$\text{Length}_i = \beta_0 + \beta_1 \text{ID}_i + \beta_2 \text{SIC}_i + \beta_3 \text{SST}_i + \beta_4 \text{Chla}_i + \beta_5 \left(\text{SST} \times \text{Chla}\right)_i + \left(1|\text{Year}_i\right) + \epsilon_i$$

where Length is the krill *Length* for observation *i*, used as a proxy for growth, and $\beta_0$ is the intercept. The fixed effects include *ID*, which represents the management unit (stratum) as a categorical variable to account for spatial heterogeneity in krill growth. Sea ice concentration (SIC), sea surface temperature (SST), and chlorophyll-a concentration (Chla) are included as environmental covariates, as they influence krill food availability and metabolic rates. An interaction term between SST and Chl is also considered to capture potential synergistic effects on krill growth. A random effect is included for *Year* to account for interannual variability in krill growth. This prevents observed trends from being biased by year-specific factors and mitigates the impacts of unbalanced sampling, which is common across the strata (ID). The residual error, $\epsilon_i$, is assumed to follow a normal distribution. This modeling approach provides a robust framework for assessing krill growth patterns while handling missing data, temporal variability, and potential correlations within the dataset. The estimated slope coefficients and associated 95% confidence intervals (CIs) indicate the strength of the relationship between variables, in which coefficients that overlap zero indicate no relationship with krill length. The variance

components indicate the amount of variation that is explained by the random factors. We computed CIs and *p-values* using a Wald *t*-distribution approximation. Along with the mean length, we also configured a model using the 75th percentile of lengths as an alternative way to model the response. In total, six linear models were established, and the selection was based on model performance. Pearson tests and the GLMM were carried out with the *lm4r* and *easystats* packages [72,78]. Analysis and manipulation of variables were grouped into strata using the *sf* and *CCAMLRGIS* packages [70,79]. A complete description of the linear models used can be found in Supporting Information 2.

## LBSPR model

LBSPR estimates spawning potential using catch length composition and life-history parameters. It has been applied to species with diverse reproductive strategies [44,80]. The method requires assumptions about growth, maturity, mortality, and selectivity by fishery taken from previous works about krill [81,82], which are listed in Table 1 in S1 File. Biological parameters included von Bertalanffy asymptotic length ($L_{inf}$), M/K ratio (natural mortality and von Bertalanffy K in a coefficient ratio), length at 50% maturity ($L_{50}$), length at 95% maturity ($L_{95}$), Coefficient of Variation of length (CVL), and maturity by length (maturity ogive). Fishery parameters include length at 50% selectivity ($SL_{50}$), length at 95% selectivity ($SL_{95}$), and the F/M ratio (fishing mortality and natural mortality). The model was evaluated by fitting it to the krill length data. Since most estimates of biological and reproduction parameters for krill have been made at regional or circumpolar scales [57], we assumed the same set of parameters for the five strata of Subarea 48.1. The methodological development from which the SPR calculation is derived was extracted from [49], where the ratio of lifetime average egg production per recruit (EPR) was calculated for fished and non-fished population. Like any assessment method, the LBSPR model relies on several of simplifying assumptions. In particular, the LBSPR model assumes population equilibrium and that the length composition data is representative of the harvested population at steady state [43,44], which we assume plausible for krill.

## Biological reference points for krill

By definition, the SPR is equal to 100% in an unexploited stock, and zero in a non-spawning stock (e.g., all mature fish have been removed, or all females have been caught). The $F_{40\%}$, that is, the fishing mortality that allows the escapement of 40% of the biomass to Maximum Sustainable Yield (MSY), is the fishing mortality rate that translates into $SPR_{40\%}$, and is used as a BRP for many species worldwide [48,83]. Insightful considerations regarding BRPs and life history parameters are available for many species [55], three distinct life-history strategies, labeled I, II, and III, which reflect differences in population growth rate (r) and carrying capacity (K) characteristics. These life-story strategies are contingent upon varying levels of reproductive productivity (high or low) and the nature of population growth (fast or slow) [84–86]. Krill is an organism with high productivity and relatively rapid growth [82], which corresponds to a Type I species (r-strategy), with M/k ~ 1 (m = 0.4, k = 0.43) therefore, it requires a higher SPR for population replenishment [44,55]. These considerations about the life strategy are useful considerations in the design of any fishery management procedure based on SPR. We used two reference points for intrinsic krill productivity, $SPR_{20\%}$ as the limit reference point and $SPR_{75\%}$ as the target for this fishery, which follows with the 20% and 75% biomass reference points used by the CCAMLR management scheme [22,23,25,28,87].

## Sensitivity scenarios analysis

Environmental factors can exert a significant influence on krill growth rates. Consequently, individual growth trajectories may serve as proxies for the environmental conditions encountered by an organism during its lifespan [59]. SPR estimates from the LBSPR model are sensitive to changes in growth parameters [44,88–92]. Considering this, we tested 10 values between the lower and upper range for $L_{inf}$ (55–65 mm) increasing by 1 mm. Secondly, we evaluated three theoretical *k* scenarios related to environmental conditions, expressed as low growth (*k* = 0.2), medium growth (*k* = 0.7, current parameter used in krill management), and high growth (*k* = 1.2) (Fig 2 in S1 File). Each scenario was applied to the five strata

within the Subarea 48.1, which comprised a total of 65 combined scenarios. Results were compared with those provided in Table 1 in S1 File serving as the base model (in particular, $k = 0.45$, $L_{inf} = 60$ mm). This analytical approach investigated how fluctuations in environmental drivers, specifically those influencing growth (parameters $L_{inf}$ and $k$) of organisms like krill, can affect the reproductive success of the species. By examining these interconnections, we aim to elucidate the mechanisms underlying krill population dynamics and potential vulnerabilities in response to future environmental changes. Model equations, figures, tables and auto-reproducible guide to application of the LBSPR model are available in S1 File.

## Results

### Environmental influences on krill growth

Pearson correlation analysis revealed that all three environmental variables had a statistically significant effect on krill population structure. A strong negative correlation was observed between chlorophyll-a and krill length ($r = -0.43$), while an even stronger negative relationship was found between Chla and SST ($r = -0.73$) (Fig 3 in S2 File). Among the evaluated linear models, *Model 5* was identified as the best-performing model, based on the coefficient of determination, root mean square error, Bayesian information criterion and Akaike information criterion among others. This model explicitly assessed the effect of spatial strata (*ID*) on krill length while incorporating key environmental covariates, including SIC, SST, and their interaction with Chla (Figs 5 and 6 in S2 File and Table 1). Although the Joinville Island (JOIN) stratum had only two years of data, its removal was not necessary, as the linear mixed model effectively accounted for missing data within grouped variables considered as random effects. Some random effects associated with *ID* were significant, indicating differences among strata. While certain strata like Gerlache and Joinville Island exhibited variations in mean length (*GS*: $\beta = 4.07$, *JOIN*: $\beta = 0.49$), the overall spatial effect remained inconsistent across regions. Additionally, the effect of SIC

**Table 1. Antarctic krill SPR estimates from CCAMLR(1) Subarea 48.1, by management strata and year (parentheses represent the standard deviation). BS = Bransfield Strait, EI = Elephant Island, GS = Gerlache Strait, JOIN = Joinville Island, SSWI = Southwest.**

| Year | BS | EI | GS | JOIN | SSWI |
|------|------|------|------|------|------|
| 2001 | 0.132 (0.024) | 0.223 (0.062) | – | – | 0.196 (0.011) |
| 2004 | 0.15 (0.008) | 0.219 (0.028) | – | – | 0.21 (0.045) |
| 2005 | 0.25 (0.014) | – | – | – | 0.253 (0.005) |
| 2006 | 0.214 (0.009) | – | 0.152 (0.09) | – | 0.349 (0.025) |
| 2007 | 0.108 (0.078) | 0.095 (0.008) | 0.069 (0.014) | – | 0.277 (0.057) |
| 2008 | – | 0.069 (0.008) | – | – | NA |
| 2009 | – | 0.322 (0.058) | – | – | 0.232 (0.016) |
| 2010 | 0.197 (0.004) | 0.329 (0.02) | 0.242 (0.014) | – | 0.31 (0.008) |
| 2011 | 0.185 (0.012) | – | – | – | 0.426 (0.009) |
| 2012 | 0.156 (0.004) | – | 0.148 (0.003) | – | 0.386 (0.006) |
| 2013 | 0.13 (0.001) | 0.191 (0.013) | 0.123 (0.002) | – | 0.176 (0.005) |
| 2014 | 0.175 (0.002) | 0.201 (0.007) | 0.172 (0.006) | – | 0.311 (0.004) |
| 2015 | 0.19 (0.003) | 0.389 (0.022) | 0.177 (0.003) | – | 0.386 (0.026) |
| 2016 | 0.288 (0.01) | 0.36 (0.057) | 0.293 (0.007) | 0.331 (0.19) | 0.266 (0.033) |
| 2017 | 0.165 (0.004) | – | 0.214 (0.016) | 0.249 (0.022) | 0.27 (0.01) |
| 2018 | 0.165 (0.003) | 0.334 (0.019) | 0.162 (0.004) | – | 0.294 (0.018) |
| 2019 | 0.161 (0.004) | 0.425 (0.02) | 0.141 (0.005) | – | 0.349 (0.006) |
| 2020 | 0.11 (0.004) | – | 0.087 (0.004) | – | 0.24 (0.012) |

(1)Commission for the Conservation of Antarctic Marine Living Resources.

was negligible ($\beta = -0.0086$). To compare the performance of models with collinearity among covariates, we systematically tested models with and without this interaction. Comparisons between models including Chla and SST independently and those incorporating their interaction term suggested a significant influence of this interaction on krill growth variability ($\beta = -0.11$, $p < 0.05$). The model designed to predict the 75th percentile of length demonstrated poor performance and was not included in the final analysis. A ranking of model performance and full description of general linear mixed models used is provided in Table 1 of S2 File.

**LBSPR model output**

The LBSPR model outputs exhibited adequate fits, as evidenced by the residual analysis of length composition distributions for krill across the strata (Figs 3 and 4 in S1 File). The model accurately captured the distribution patterns of size classes, indicating its effectiveness in characterizing the population structure and variations in length distribution due to the natural variability in krill populations between years and strata. Comparisons between the simulated length distributions of unfished populations with the observed length distributions within each stratum showed that Southwest, Gerlache and Elephant Island strata exhibited the greatest differences from the simulated structure. Conversely, the Bransfield Strait and Joinville Island strata demonstrated the empirical data were closer to the virginal condition simulated by the model (Fig 4 in S1 File). Bransfield stratum stood out as having a lower proportion of mature individuals, suggesting a higher prevalence of juveniles. It is important to note that the same maturity parameters were applied across all strata (Fig 5 in S1 File).

Krill SPR varied significantly across years and strata. Notably, none of the strata yielded SPR above the reference level of 75% in any year examined, and all strata, with the exception of Joinville (only 2 years of data) and the last years of the series in Elephant Island and Southwest, contained years with SPR below the limit reference point of 20% (Fig 2). In fact, intrinsic productivity in Bransfield and Gerlache strata remained consistently below $SPR_{20\%}$ for all years in the time series, while Elephant Island and Southwest showed an increasing trend in SPR, which corresponded to the increased prevalence of adults in the catch in recent years (Fig 2). All the estimated values and their associated standard deviation of SPR by stratum and by year are shown in Table 1. The analysis for the whole of Subarea 48.1 indicated that SPR values remain stable over time, predominantly below the 0.25 threshold, with no significant trends observed (Table 5 in S1 File).

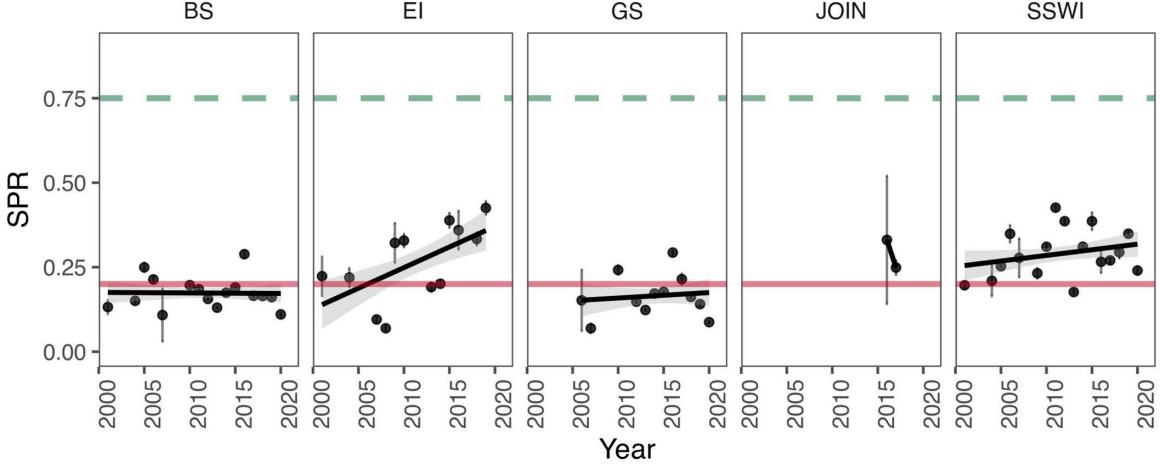

**Fig 2. Krill intrinsic productivity (SPR) by management strata and by year.** The green dashed line represents 75% SPR (reference target) and the red line is 20% SPR (reference limit). BS = Bransfield Strait, EI = Elephant Island, GS = Gerlache Strait, JOIN = Joinville Island, SSWI = Southwest.

## Sensitivity analysis

We found that as asymptotic length ($L_{inf}$) increased, the median SPR values generally decreased, with the highest medians observed at $L_{inf} = 55$ mm and the lowest at $L_{inf} = 65$ mm. Specifically, the median SPR values for the Bransfield Strait ranged from 0.28 at $L_{inf} = 55$ mm to 0.12 at $L_{inf} = 65$ mm. In general, lower $L_{inf}$ values in the LBSPR model resulted in higher SPR estimates, a trend that was consistent across all strata. These results suggest a tendency toward greater stability in SPR estimates as $L_{inf}$ increases (Fig 3 and Table 2). Regarding the three growth rates ($k$) tested (low, medium, and high), high and medium growth rates produced very low SPR estimates compared to low growth, with particularly high individual growth leading to SPR values very close to the reference level of 75% SPR, while remaining far from the limit reference level of 20% (Fig 4 and Table 2).

Sensitivity analysis of the asymptotic length ($L_{inf}$) for Subarea 48.1, shows that SPR estimates consistently fall below the target reference point of 0.75, although some variability occurs across the tested parameter range, with values clustering around the 0.20 threshold. In contrast, sensitivity to the $k$ reveals a strong influence on SPR outcomes: the "Low" growth scenario results in higher SPR values (~0.6), whereas the "Med" and "High" growth scenarios yield significantly lower values, consistently below 0.2. Overall, the analysis points to low spawning potential for krill in area 48.1, with results being robust to $L_{inf}$ but highly sensitive to growth rate assumptions (Fig 5).

## Discussion

### Spatial and temporal heterogeneity in intrinsic krill productivity

Spatial heterogeneity is common in marine populations [93]. Population dynamics and changes in spatial patterns of krill productivity can be influenced by fishing activity, or the fishing activity follows those changes, concentrating in places where the population is most productive. Changes in the ecosystem triggered by environmental and ecological variables are also direct drivers that cause changes in the population structure of krill on the spatial scale considered here [4,17,22]. Consequently, it seems imperative to account for spatial heterogeneity in stock structure and distribution when evaluating stock indicators. This is particularly critical for krill, as the fishery has transitioned from a widely dispersed to a highly

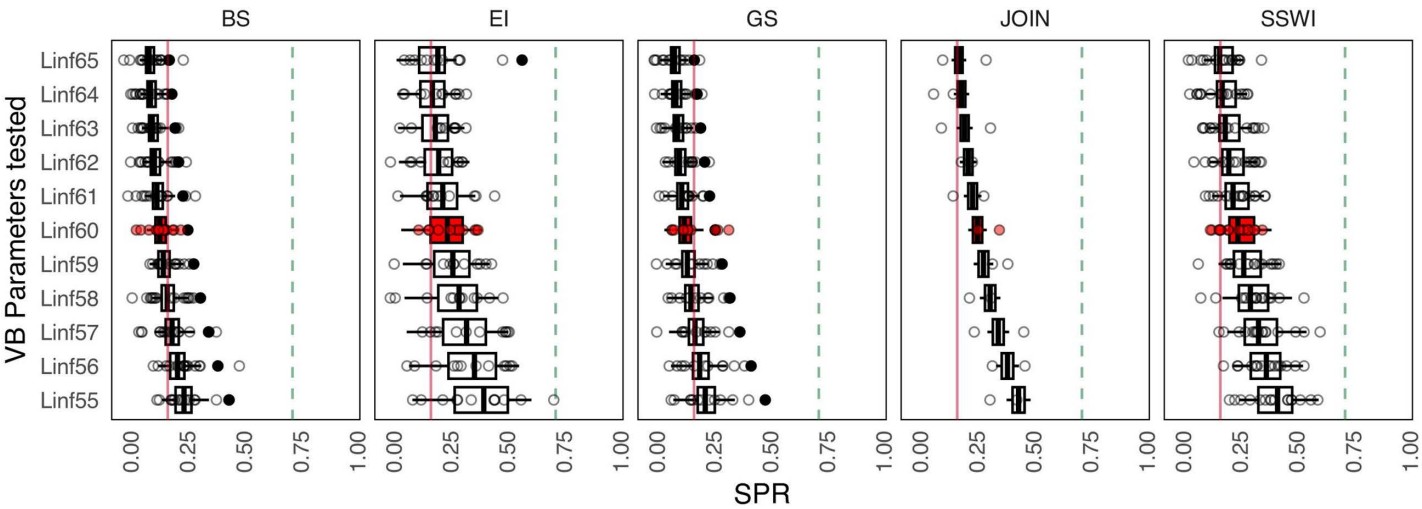

**Fig 3. Sensitivity analysis in SPR estimation by management strata in relation to the scenario of a range of Von Bertalanffy asymptotic length values for Antarctic krill.** The green dashed line represents 75% SPR (reference target), and the red dashed line is 20% SPR (reference limit). Red boxes represent current parameters used in krill management. BS = Bransfield Strait, EI = Elephant Island, GS = Gerlache Strait, JOIN = Joinville Island, SSWI = Southwest.

**Table 2. Antarctic krill LBSPR estimates from CCAMLR Subarea 48.1 by management strata in relation to a range of theoretical krill growth parameters, including eleven asymptotic lengths and three growth rate coefficients (low = 0.2, medium = 0.7, and high = 1.2) (parentheses represent the standard deviation). In bold, the values of the parameters currently used in management (Maschette et al, 2020). BS = Bransfield Strait, EI = Elephant Island, GS = Gerlache Strait, JOIN = Joinville Island, SSWI = Southwest.**

| | | BS | EI | GS | JOIN | SSWI |
|---|---|---|---|---|---|---|
| Asymptotic length ($L_{inf}$) | Linf55 | 0.28 (0.07) | 0.46 (0.23) | 0.27 (0.11) | 0.47 (0.07) | 0.47 (0.13) |
| | Linf56 | 0.25 (0.07) | 0.42 (0.23) | 0.24 (0.09) | 0.42 (0.07) | 0.42 (0.11) |
| | Linf57 | 0.23 (0.06) | 0.39 (0.23) | 0.22 (0.08) | 0.38 (0.07) | 0.38 (0.1) |
| | Linf58 | 0.21 (0.06) | 0.36 (0.23) | 0.2 (0.08) | 0.35 (0.06) | 0.34 (0.09) |
| | Linf59 | 0.19 (0.05) | 0.34 (0.23) | 0.18 (0.07) | 0.32 (0.06) | 0.32 (0.08) |
| | Linf60 | **0.17 (0.05)** | **0.32 (0.23)** | **0.17 (0.06)** | **0.29 (0.06)** | **0.29 (0.07)** |
| | Linf61 | 0.16 (0.04) | 0.3 (0.21) | 0.15 (0.06) | 0.27 (0.06) | 0.27 (0.07) |
| | Linf62 | 0.15 (0.04) | 0.27 (0.18) | 0.14 (0.05) | 0.25 (0.05) | 0.25 (0.06) |
| | Linf63 | 0.14 (0.04) | 0.25 (0.17) | 0.13 (0.05) | 0.23 (0.05) | 0.23 (0.06) |
| | Linf64 | 0.13 (0.04) | 0.24 (0.15) | 0.13 (0.05) | 0.22 (0.05) | 0.22 (0.05) |
| | Linf65 | 0.12 (0.04) | 0.22 (0.14) | 0.12 (0.04) | 0.21 (0.05) | 0.21 (0.05) |
| Growth Scenario (k) | Low | 0.49 (0.08) | 0.61 (0.19) | 0.47 (0.14) | 0.71 (0.05) | 0.65 (0.11) |
| | Med | **0.08 (0.03)** | **0.17 (0.13)** | **0.08 (0.03)** | **0.15 (0.05)** | **0.15 (0.05)** |
| | High | 0.05 (0.02) | 0.1 (0.08) | 0.04 (0.02) | 0.08 (0.03) | 0.09 (0.03) |

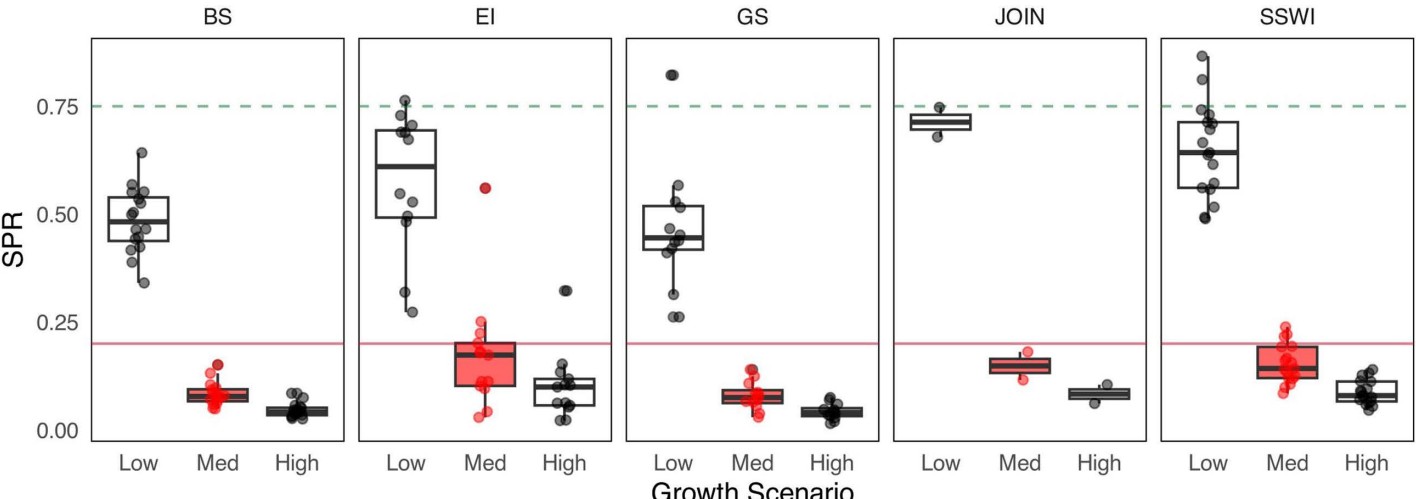

**Fig 4. Sensitivity analysis to SPR estimation by management strata in relation to a range of growth coefficients (k) representing three different Antarctic krill growth rates (low = 0.2, medium = 0.7, and high = 1.2).** The green dashed line represents 75% SPR (reference target) and the red dashed line is 20% SPR (reference limit). Red boxes represent current parameters used in krill management. BS = Bransfield Strait, EI = Elephant Island, GS = Gerlache Strait, JOIN = Joinville Island, SSWI = Southwest.

concentrated spatial distribution into small areas over the past 20 years, particularly in Subarea 48.1. Such concentration may amplify the potential impacts of harvesting. To date, a quantitative assessment of spatial heterogeneity in stock status for krill has been lacking, hindering effective management [4,14,15,26,94].

Our analysis of length compositions from the krill fishery reveals significant spatial heterogeneity in population structure and its temporal changes. Length compositions of the catches were used as they are widely regarded as one of the most representative sources of information on the population dynamics of exploited marine resources

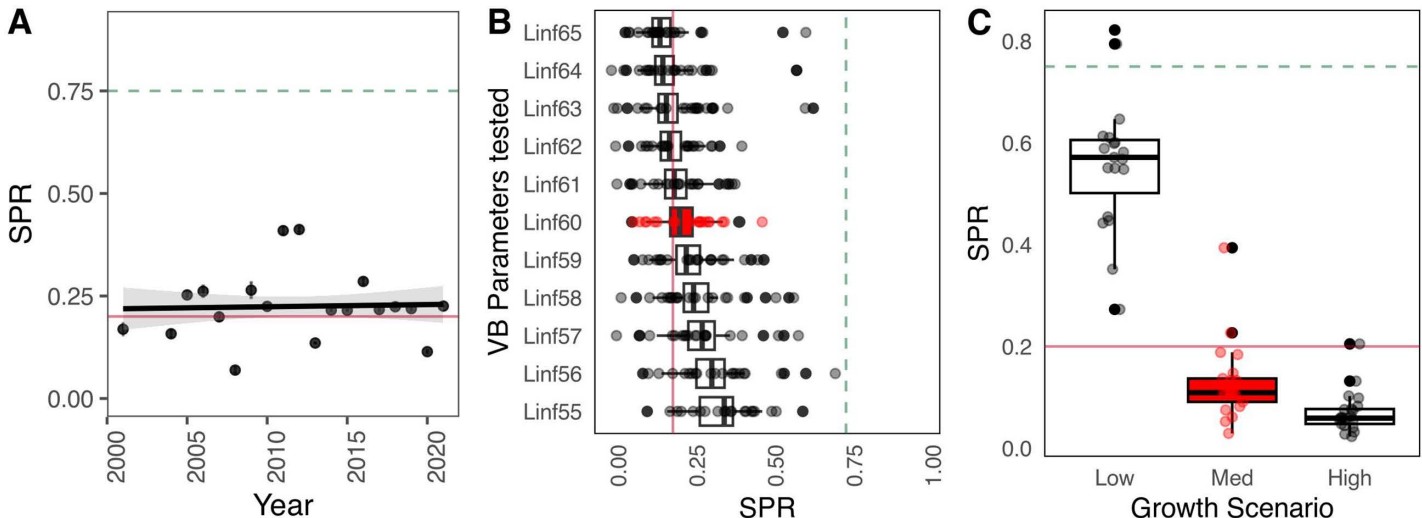

**Fig 5. Overall results of SPR estimation and sensitivity scenarios of Antarctic krill for Subarea 48.1. A)** SPR values in time series (2000-2020), **B)** scenario comparisons with VB asymptotic length values and **C)** SPR variability for growth rates scenario.

[41,46,52]. These data provide valuable insights as an indicator of population status [41], inform biomass estimates [95], and serve as a key component in integrated stock assessment models [96]. Changes in the availability, distribution, and concentration of harvested marine populations such as krill have been reflected in length-composition data, and such data and size-based models have been widely used to make recommendations for sustainable fishery management [14,41,44,47,50,53].

We verified that the intrinsic productivity of krill varied across management strata and changed over time. Elephant Island and the Southwest South Shetland Islands exhibited the highest SPR levels, showing a positive trend in the latter years of the series. In contrast, the Bransfield Strait and Gerlache strata consistently displayed low SPR levels throughout the entire time series. These strata are characterized by low productivity and a relatively high abundance of immature juvenile individuals, as highlighted in studies by [12] and [38]. Given that krill fishery activity has expanded to these strata, it seems useful to prioritize these strata for effective management, especially because these strata also serve as a recurrent foraging areas for krill predators, including baleen whales [97,98]. In Bransfield strata, the SPR has decreased throughout the study period, with the lowest level of 12% SPR occurring in 2020. This may be related to the growing fishing activity that has increased and concentrated in this strata [10,99], which has targeted older krill, thus, reducing reproductive capacity of the population and consequently, decreasing intrinsic productivity. This spatial and temporal heterogeneity of SPR of krill provides a biological mechanism for the consequential changes observed in indicators of productivity across the WAP [16,17]. From a holistic point of view, these spatiotemporal differences in krill SPR are likely to significantly impact the maintenance of long-term ecosystem balance and species diversity [4,22]. Higher SPR ensures a larger number of offspring, increasing the chances of population recovery after disturbances or environmental fluctuations, while reduced SPR may cause decreased food availability for dependent predators [20]; hence, fluctuations in intrinsic productivity can have cascading effects on krill population and their predators. Spatial heterogeneity in the population structure of krill (demographic patterns, recruitment) has been widely described for different areas of the SO, specifically in the WAP [11,12,14]. However, these studies are mostly descriptive. In contrast, the LBSPR approach is a model-based method that facilitates a quantitative assessment of the spatio-temporal differences in the krill population structure that can be used to provide meaningful spatially-explicit recommendations for conservation and management, consistent with CCAMLR's mandate for ecosystem-based fishery management.

Our results support those from previous length-based analyses about growth [42,46,47], but it is the first time that such an analysis has been carried out on krill. In this context, two aspects emerge. Firstly, despite recognizing spatial variability in demographic structures among strata, we employed the same set of life history parameters and assumptions for our analysis. Secondly, we identified an information gap regarding biological parameters in krill, crucial for both our analysis and stock assessment generally. Consequently, conducting studies to determine specific life history and maturity parameters by stratum becomes imperative to refine understanding. Nevertheless, sensitivity analysis could serve as a valuable tool to address these existing knowledge gaps.

## Environmental impacts to krill intrinsic productivity

Krill population dynamics vary in distribution, biomass, recruitment, and phenology. The main drivers of these changes are related to different environmental variables in the WAP [8,9,14,18,19,21,38,69]. While acknowledging the influence of environmental factors on variables such as biomass and recruitment [12,18,100], it remains imperative to systematically assess spatial and temporal shifts and their impacts on krill intrinsic productivity. Our findings indicate a strong, negative relationship between chlorophyll-a concentration and krill size, suggesting that environmental conditions significantly influence krill population structure, which is in accordance with previous research [8,18,21,59,101,102]. Previous research has highlighted the temporal variability in krill population characteristics [12,103]. Similarly, spatial components (strata) and environmental variables, like interaction between Chla and SST contribute meaningfully to explaining length variation in all of the management strata considered here. Increased chlorophyll-a levels have been linked to enhanced krill recruitment [6,18], but paradoxically, this can also lead to reductions in intrinsic productivity as the local population is dominated by immature individuals. This broader perspective may explain discrepancies with previous analysis. While [102] established a positive correlation between female maturity and krill recruitment, our study expanded this analysis by assessing the spawning potential of the entire population, including both males and females. However, understanding the interplay between fishing activities and environmental conditions on krill physiology and reproduction remains a significant challenge in the face of climate change.

A good practice in the use of stock assessment models, involves testing scenarios to determine the impact of uncertainty sources, like growth parameters, on the output, detects biases and improve model reliability [52,104]. Indeed, few studies have performed parameter sensitivity analyses for these methods with real case studies [41,42,44,105]. Given the strong negative influence of chlorophyll-a on krill growth in the WAP (Fig 3 in S2 File), we performed krill SPR sensitivity analyses across various values of growth parameters. Recognizing that the parameters of the von Bertalanffy growth curve are correlated [106–108], we applied the analytical derivation by [44] to calculate SPR and independently tested scenarios for each parameter to evaluate their effects.

Testing different von Bertalanffy growth parameter scenarios also revealed that slower growth rates (e.g., $k=0.2$) result in higher SPR values across all strata, indicating greater resilience to environmental and anthropogenic changes (Fig 4), consistent with [43], there is a higher level of resilience to anthropogenic and environmental changes. Conversely, faster growth rates were associated with lower SPR values, potentially underestimating the risks of overfishing. Overestimating $k$ may underestimate SPR, falsely suggesting stock sustainability, while underestimating $k$ could lead to overestimated SPR values, providing a false sense of security. These findings underscore the importance of precise, stock-specific estimates of life-history parameters, as misestimations can lead to inappropriate regulatory measures and jeopardize long-term sustainability. [105] and [109] recommend deriving life-history parameters, such as $L_{inf}$ and $k$, from stock-specific studies to improve the accuracy of length-based assessment methods like LBSPR. In the case of Antarctic krill, while the existing stock assessment model (e.g., the Grym; [81]) incorporates life-history parameters, these data are often outdated. Most research focuses on physiological and reproductive aspects, with limited emphasis on parameters directly used in LBSPR. Addressing these gaps is critical and could involve utilizing available data and estimating missing parameters through other methods [e.g., 74]. A quantitative tool like LBSPR, which can detect changes in krill productivity, offers valuable

insights into the overall health and resilience of the WAP ecosystem and its krill populations, can play a useful role in informing sustainable management strategies.

## CCAMLR krill fishery management and spatial heterogeneity

A precautionary management approach for the krill fishery would seem to rely on maintaining a healthy population through sustainable harvest policy, which can be influenced by variations in intrinsic productivity. In a standard fishery science context, a robust management procedure typically includes a harvest control rule (HCR), which is based on an algorithm through which the mortality (harvest) rate or quota to be implemented is determined [30,110]. The HCR provides for adjusting catch limits in accordance with a target fishery reference point (fishing mortality or biomass) that relates to changes in an indicator of stock status, such as SPR. The current krill fishery harvest strategy in Subarea 48.1 is based on a constant catch that is not related to changes in a population or ecosystem indicator or associated target fishery reference point, such as Maximum Sustainable Yield (MSY), i.e., maintaining biomass at sustainable levels. CCAMLR has recognized these gaps [13,22,25,26], particularly the absence of a stock assessment process and the lack of consideration for spatial components in the current management recommendations. A new management strategy for the krill fishery has been in development over the last 5 years [3,22,28,36]. The new strategy includes three pivotal elements: i) regularly updated estimates of krill biomass, whether at regional or small scales; ii) a stock assessment approach to estimate precautionary catch limits; and iii) an overlap analysis that considers krill-dependent predators for the spatial allocation of catch limits among smaller management units within the region [13,35]. Despite these proposals for assessing krill population dynamics at a finer level, these three elements are not yet operational for decision making [3,26]. In fact, in 2024, the lack of consensus on the implementation of the revised management strategy resulted in the expiration of a conservation measure (CM 51−07) that allocated catches among subareas

The expiration of CM 51−07 highlights the need for an updated spatial management strategy. Our results show that SPR varies among strata within subareas, suggesting that a simple harvest control rule could adjust catch limits according to local reproductive potential, using reference levels such as SPR$_{75\%}$, this approach provides for a management trigger, such as changes to the harvest rate or adjusting catch limits by strata. The management procedure adjustment presented in this study identified stratum-specific differences in SPR, and a new HCR for the krill fishery could change the harvest level (catch limit) according to the levels of SPR. We can envision an adaptation to the common "hockey stick" HCR where;

$$F = \begin{cases} F_{MSY} & \text{if } \frac{SPR_{i,j}}{SPR_{MSY}} \geq 0.75 \\ F_{MSY}\left(\frac{SPR_{i,j}}{SPR_{MSY}} - 0.2\right)/0.75 & \text{if } 0.2 \leq \frac{SPR_{i,j}}{SPR_{MSY}} < 0.75 \\ 0 & \text{if } \frac{SPR_{i,j}}{SPR_{MSY}} < 0.2 \end{cases}$$

where $i$ and $j$ indicate the SPR calculated by stratum and year, and SPR$_{MSY}$ represent references in SPR$_{75\%}$, which is a common BRP to all stratum. SPR calculation scheme can be easily updated as new data becomes available. The proposal for spatial adjustment and allocation of krill catch limits is not new and has considered different criteria including the catch history, predator demand and krill standing stock within so-called Small Scale Management Unit (SSMU) [24,36,111]. It should be noted that our proposed harvest control rule and spatial allocation scheme is based on an assessment of a single stock of krill, providing a tactical, operational, quantifiable management procedure for krill in Antarctica [30,112]. It also constitutes a biologically-based allocation of spatial catches on krill for the management strata defined in [87].

While our study highlights the utility of the LBSPR method in assessing the intrinsic productivity of krill and its potential as a management tool, it is important to acknowledge the massive data available on krill populations. The LBSPR should be viewed as a complementary tool that can provide valuable insights into spatial and temporal variations in krill

productivity, especially in the interim period until an integrated stock assessment for krill is fully developed and agreed upon by CCAMLR. This integration of LBSPR into current management practices could bridge gaps in understanding and enhance the effectiveness of krill fishery management, facilitating decisions that are based on the best available science until more comprehensive stock assessment frameworks are established.

### Limitations and caveats of length-based assessment methods

Length-based assessment methods used to determine stock status are strongly governed by the biological parameters on which they depend [43,44,53]. In this study, we demonstrate that variation in growth parameters, and specifically in the von Bertalanffy growth function, substantially influences SPR estimates, as shown in the sensitivity analyses. However, the inherent structural limitations of length-based approaches such as LBSPR introduce uncertainty and bias that must be carefully considered, particularly when estimating fishing mortality and reproductive potential. One important limitation is the uneven availability of length–frequency data across years, which prevents the derivation of a consistent and continuous temporal diagnostics of population status. This issue is especially evident in poorly monitored areas such as the Joinville Island stratum, where only two years of data are available (2016–2017), restricting inference to those specific years. Second, LBSPR is sensitive to the shape of the length distribution, and biases may arise when samples disproportionately represent juveniles or adults. An overrepresentation of adults may lead to an underestimation of the fishing impact on the mature fraction of the population, thereby inflating SPR estimates and masking true reproductive depletion, a pattern also noted by [80] and [105]. Although data from the krill fishery generally capture a broad size range, spatial heterogeneity can lead to natural over-representation of certain life stages, such as the strong proportion of juveniles in the Bransfield Strait, which may influence LBSPR outputs despite reflecting biological rather than sampling driven patterns. Third, length-based models tends to overestimate the SPR when stocks are heavily exploited, thereby amplifying bias under high exploitation rates. Those structural patterns have been widely documented. [44] showed that the performance of the LBSPR performance is highly sensitive to misspecification of life-history parameters and exploitation rates. [52] demonstrated that bias is driven by uncertainty in life history, recruitment dynamics, and exploitation status. [41] found that reduced sample size and exploitation rates further exacerbate these biases. [113] noted that length-based approaches may outperform catch-based methods in some situations, but also, at high exploitation levels, that the LBSPR and similar estimators may become biased and imprecise. These structural concerns may be influencing the SPR estimates obtained in this study. However, the objective here is not to deliver a definitive stock diagnosis, but to generate a biologically meaningful and spatially explicit indicator of intrinsic productivity (e.g., SPR) that can inform improved spatial management procedures for Antarctic krill.

### Conclusion

This study undertakes the first analysis of the intrinsic productivity of krill in Subarea 48.1 using Length-Based Spawning Potential Ratio (LBSPR) method. This quantitative approach proved to be a valuable tool for assessing the influence of multiple stressors on the spatial variability of krill intrinsic productivity while elucidating the environmental drivers shaping population dynamics. It also provided a robust framework for quantifying temporal trends in spatially explicit population status relative to biological reference points. These spatio-temporal variations in intrinsic productivity offer a fundamental basis for a developing a management procedure that incorporates the complexities of krill population dynamics within a framework aligned with CCAMLR's revised management strategy for Subarea 48.1. In this context, a harvest control rule that explicitly recognizes spatial differences in intrinsic productivity, together with a more biologically grounded allocation of catch across strata in the WAP, would provide a solid foundation for guiding a more appropriate distribution of catch limits in the region. By aligning fishing practices with the natural patterns of krill productivity, this approach lays the groundwork for a more balanced coexistence between harvesting activities and krill-dependents predators, ultimately contributing to the sustainability of this essential component of the Southern Ocean.

## Supporting information

**S1 File. Key formulas, figures, and code snippets for applying the LBSPR model to study krill populations in the West Antarctic Peninsula (WAP) region can be found at this link: LBSPRKrill.**
(PDF)

**S2 File. Code and analysis for identifying environmental influences on krill length using correlation and mixed-effects models across spatial and temporal scales can be found at this link: Krill Length Correlation** .
(PDF)

## Acknowledgments

The authors would like to thank the Secretariat and members of the Commission for the Conservation of Antarctic Marine Living Resources (CCAMLR) for providing access to the krill fishery data, which was instrumental to this study. We are also grateful for the institutional and logistical support, as well as the opportunity to engage in constructive discussions that allowed us to improve and refine our analyses.

## Author contributions

**Conceptualization:** George Watters, Cesar Cardenas.

**Formal analysis:** Mauricio Mardones.

**Methodology:** Mauricio Mardones, Cesar Cardenas.

**Supervision:** Cesar Cardenas.

**Visualization:** Mauricio Mardones.

**Writing – original draft:** Mauricio Mardones, Erica Jarvis Mason, Cesar Cardenas.

**Writing – review & editing:** Mauricio Mardones, Erica Jarvis Mason, Francisco Santa Cruz, George Watters, Cesar Cardenas.

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
