## [Decision Letter · Decision Letter 0]

23 May 2025

Dear Dr. MARDONES,

Thank you for submitting your manuscript to PLOS ONE. After careful consideration, we feel that it has merit but does not fully meet PLOS ONE’s publication criteria as it currently stands. Therefore, we invite you to submit a revised version of the manuscript that addresses the points raised during the review process.

We look forward to receiving your revised manuscript.

Kind regards,

Sarah Jose, Ph.D.

Staff Editor

PLOS ONE

Journal Requirements:

2. Thank you for stating the following financial disclosure: [CC; FS: INACH “Marine Protected Areas” Program (Grant No. 2409052). CC: ANID/Millennium Science Initiative Program (Grant No. ICN2021_002). MM: Doctorate Scholarship from ANID/Centro IDEAL, FONDAP at the Universidad Austral de Chile. (Grant No. 15150003) MM: Commission for the Conservation of Antarctic Marine Living Resources (CCAMLR) Scholarship Scheme (2023-2024)]. 

3. Thank you for stating the following in the Acknowledgments Section of your manuscript: [This research was supported by the following funding sources: the INACH “Marine Protected Areas” Program (Grant No. 2409052), the ANID/Millennium Science Initiative Program (Grant No. ICN2021_002), the CCAMLR Scholarship Scheme (2023-2024), and the Doctorate Scholarship from CENTRO-IDEAL at the Universidad Austral de Chile. The authors extend their gratitude to the Secretariat of the Commission for the Conservation of Antarctic Marine Living Resources (CCAMLR) for providing access to the krill fishery data, which was instrumental to this study.]

Please remove any funding-related text from the manuscript and let us know how you would like to update your Funding Statement. Currently, your Funding Statement reads as follows: [CC; FS: INACH “Marine Protected Areas” Program (Grant No. 2409052). CC: ANID/Millennium Science Initiative Program (Grant No. ICN2021_002). MM: Doctorate Scholarship from ANID/Centro IDEAL, FONDAP at the Universidad Austral de Chile. (Grant No. 15150003) MM: Commission for the Conservation of Antarctic Marine Living Resources (CCAMLR) Scholarship Scheme (2023-2024)].

4. In the online submission form, you indicated that [The original data used in this study are available through the Commission for the Conservation of Antarctic Marine Living Resources (CCAMLR) and can be obtained upon request by contacting the CCAMLR Data Centre.].

5. Please upload a copy of Supporting Information Figure/Table/etc. Supporting Information 1 and 2 which you refer to in your text on page 36.

Reviewers' comments:

Reviewer's Responses to Questions

**Comments to the Author**

1. Is the manuscript technically sound, and do the data support the conclusions?

Reviewer #1: Yes

2. Has the statistical analysis been performed appropriately and rigorously?

Reviewer #1: Yes

3. Have the authors made all data underlying the findings in their manuscript fully available?

Reviewer #1: Yes

4. Is the manuscript presented in an intelligible fashion and written in standard English?

Reviewer #1: Yes

Reviewer #1: This research is particularly interesting because it explores the use of LBSPR spatially. The manuscript is well written and presented. However, there are some areas that still need improvement to enhance the quality of this paper.

1. Adding an insert map to make it easier for the reader to know the location of the research and others that can be seen in the comments section in the reviewed manuscript file. - line 154 (Fig 1)

2. Specify the length type used. Is it carapace/total/body length? - line 162

3. Use the growth parameter symbol k (lower case) consistently to differentiate from carrying capacity (K) as well as natural mortality as M (capital) - line 230-231 and 253.

4. Please re-check the appropriate location as commented in line 324 and 327.

5. Line 331 - which one did you uses as LRP 0.25 or 0.2?

Please see the attached reviewed manuscript.

**Do you want your identity to be public for this peer review?** For information about this choice, including consent withdrawal, please see our Privacy Policy

Reviewer #1: No

---

## [Author Response · Author response to Decision Letter 1]

5 Jun 2025

We would like to express our sincere appreciation for your thoughtful review of our manuscript titled "Disparate estimates of intrinsic productivity for Antarctic krill across small spatial scales, under a rapidly changing ocean."

Thank you for your positive overall assessment and for highlighting the scientific value of applying LBSPR spatially. We are grateful for your detailed suggestions, which have helped us improve the clarity and rigor of our manuscript.

We have carefully addressed each of your comments:

We have included an inset map to facilitate spatial orientation for the reader (Figure 1).

We now specify the type of length measurement used (total length).

The growth parameter symbol k is now consistently lowercase, and M is consistently capitalized to avoid confusion.

We reviewed and corrected the location references as per your comments on lines 324 and 327.

We clarified which LRP threshold was used (0.25), with justification included in the Methods section.

We believe these revisions have strengthened our manuscript, and we are grateful for your role in improving its quality.

Editor Observations and Responses

Obs 1: When submitting your revision, we need you to address these additional requirements.

Reply: Fixed Author and Affiliation issues in “Revised Manuscript with Track Changes” and “Manuscript” regarding guidelines

Obs 2: Thank you for stating the following financial disclosure: [CC; FS: INACH “Marine Protected Areas” Program (Grant No. 2409052). CC: ANID/Millennium Science Initiative Program (Grant No. ICN2021_002). MM: Doctorate Scholarship from ANID/Centro IDEAL, FONDAP at the Universidad Austral de Chile. (Grant No. 15150003) MM: Commission for the Conservation of Antarctic Marine Living Resources (CCAMLR) Scholarship Scheme (2023-2024)].

Reply: We appreciate the observation regarding the role of the funders. We confirm that we will include the following statement in our revised cover letter:

Please proceed with updating the online submission form accordingly.

Obs 3: Thank you for stating the following in the Acknowledgments Section of your manuscript: [This research was supported by the following funding sources: the INACH “Marine Protected Areas” Program (Grant No. 2409052), the ANID/Millennium Science Initiative Program (Grant No. ICN2021_002), the CCAMLR Scholarship Scheme (2023-2024), and the Doctorate Scholarship from CENTRO-IDEAL at the Universidad Austral de Chile. The authors extend their gratitude to the Secretariat of the Commission for the Conservation of Antarctic Marine Living Resources (CCAMLR) for providing access to the krill fishery data, which was instrumental to this study.]

Please remove any funding-related text from the manuscript and let us know how you would like to update your Funding Statement. Currently, your Funding Statement reads as follows: [CC; FS: INACH “Marine Protected Areas” Program (Grant No. 2409052). CC: ANID/Millennium Science Initiative Program (Grant No. ICN2021_002). MM: Doctorate Scholarship from ANID/Centro IDEAL, FONDAP at the Universidad Austral de Chile. (Grant No. 15150003) MM: Commission for the Conservation of Antarctic Marine Living Resources (CCAMLR) Scholarship Scheme (2023-2024)].

Reply: We acknowledge the observation and confirm that we have removed all funding-related information from the Acknowledgments section of the new manuscript, in accordance with the editor's recommendation. Only the expressions of gratitude remain, as suggested.

Additionally, we have incorporated the complete funding information in the revised cover letter, as requested. Please proceed with updating the Funding Statement section of the online submission form accordingly.

Obs 4: In the online submission form, you indicated that [The original data used in this study are available through the Commission for the Conservation of Antarctic Marine Living Resources (CCAMLR) and can be obtained upon request by contacting the CCAMLR Data Centre.].

Reply: We appreciate the opportunity to clarify our Data Availability statement.

The original data used in this study are managed by the Commission for the Conservation of Antarctic Marine Living Resources (CCAMLR), which applies strict policies for sharing data generated within the jurisdiction of its Statistical Subareas. These restrictions are in place due to the rules of access and use of CCAMLR Data as agreed by it Member States. As a result, we are not permitted to publicly share the original raw data.

However, to ensure full reproducibility of our analyses, we have provided cleaned and anonymized data templates, along with all relevant R code, in the following publicly accessible GitHub repository: https://github.com/MauroMardones/LBSPR_Krill

This repository is also linked in the Supporting Information 1 under the "Code Repository" section. We believe this solution allows other researchers to fully replicate the analytical workflow and verify the results.

We thank you for your understanding regarding these data-sharing constraints.

Obs 5: Please upload a copy of Supporting Information Figure/Table/etc. Supporting Information 1 and 2 which you refer to in your text on page 36.

Reply:

Supporting Information 1 and 2 were originally provided during the initial submission as web links to the author's GitHub page, as they are part of the project’s reproducible workflow. Although these links remain publicly accessible and are cited accordingly in the Supporting Information section in manuscript (page 36), we have now also uploaded Supporting Information 1 and 2 in PDF format to the submission portal, as requested by the editor. This ensures the material is available both through the manuscript references and directly via the journal platform.

Obs 6: Please review your reference list to ensure that it is complete and correct. If you have cited papers that have been retracted, please include the rationale for doing so in the manuscript text, or remove these references and replace them with relevant current references. Any changes to the reference list should be mentioned in the rebuttal letter that accompanies your revised manuscript. If you need to cite a retracted article, indicate the article’s retracted status in the References list and also include a citation and full reference for the retraction notice.

Reply:

We have reviewed and updated the formatting of these references without altering their position in the manuscript.

25. Hill S, Hinke J, Ratcliffe N, Trathan P, Watters G. WG-EMM-2019/28 Advances are urgently needed in providing regular estimates of krill stock status based on the available data. CCAMLR; 2019.

40. Molles MC, Sher A. Ecology: Concepts and applications. 8th ed. New York, NY: McGraw-Hill Education; 2019.

56. Mace PM. A new role for MSY in single-species and ecosystem approaches to fisheries stock assessment and management. Fish and Fisheries. 2001;2: 2–32. doi:10.1046/j.1467-2979.2001.00033.x

76. Maschette D, Wotherspoon S, Kawaguchi S, Ziegler P. WG-FSA-2021/39 29. Grym assessment for Subarea 48.1 Euphausia superba populations. CCAMLR; 2021.

Reviewer 1 Observations and Replys

Reviewer #1: This research is particularly interesting because it explores the use of LBSPR spatially. The manuscript is well written and presented. However, there are some areas that still need improvement to enhance the quality of this paper.

Obs 1: Adding an insert map to make it easier for the reader to know the location of the research and others that can be seen in the comments section in the reviewed manuscript file. - line 154 (Fig 1)

Reply: We have now added a new Fig 1.

Obs 2: Specify the length type used. Is it carapace/total/body length? - line 162

Reply: Detailed in lines 178-180;

“…International Scientific Observation (SISO). Krill length compositions (total length (TL) measured from the anterior edge of the eye to the posterior end of the telson, excluding terminal spines)…”

Obs 3: Use the growth parameter symbol k (lower case) consistently to differentiate from carrying capacity (K) as well as natural mortality as M (capital) - line 230-231 and 253.

Reply: We have now corrected this throughout the manuscript, distinguishing between K (carrying capacity) and k (von Bertalanffy growth coefficient), and capitalizing natural mortality (M) accordingly yin all document.

Obs 4: Please re-check the appropriate location as commented in line 324 and 327.

Reply: Thanks for the observation. The reviewer is correct—strata South West were properly added with the characteristics described in the corresponding paragraph.

5. Line 331 - which one did you uses as LRP 0.25 or 0.2?

Reply: Thanks for pointing this out. We can confirm that the value used as the Limit Reference Point (LRP) was 0.20. The manuscript has been corrected accordingly.

---

## [Decision Letter · Decision Letter 1]

31 Oct 2025

Dear Dr. MARDONES,

Thank you for submitting your manuscript to PLOS ONE. After careful consideration, we feel that it has merit but does not fully meet PLOS ONE’s publication criteria as it currently stands. Therefore, we invite you to submit a revised version of the manuscript that addresses the points raised during the review process.

I apologize for the delay in communicating this decision. After the initial reviews, the manuscript was leaning toward a minor revision. However, a second reviewer subsequently provided detailed feedback indicating that Major Revisions are necessary. Upon careful consideration of all reports, it is clear that the issues raised by Reviewer 2 are substantial and require significant effort. Therefore, I'm changing the official decision to Major Revision.

Specifically, the reviewers indicate you need to deeply clarify your model, provide a more rigorous discussion of data limitations, and do a substantial restructuring and simplification of the language.

The detailed comments from Reviewer 2 are below. Please address each point thoroughly in your revised manuscript and point-by-point response document.

We look forward to receiving your revised manuscript.

Kind regards,

Clara F. Rodrigues

Academic Editor

PLOS ONE

Journal Requirements:

Reviewers' comments:

Reviewer's Responses to Questions

**Comments to the Author**

Reviewer #1: All comments have been addressed

Reviewer #2: (No Response)

2. Is the manuscript technically sound, and do the data support the conclusions?

Reviewer #1: Yes

Reviewer #2: Partly

3. Has the statistical analysis been performed appropriately and rigorously?

Reviewer #1: Yes

Reviewer #2: I Don't Know

4. Have the authors made all data underlying the findings in their manuscript fully available?

Reviewer #1: Yes

Reviewer #2: No

5. Is the manuscript presented in an intelligible fashion and written in standard English?

Reviewer #1: Yes

Reviewer #2: No

Reviewer #1: All my comments have been addressed well. From my side, this manuscript was ready for publication.

Reviewer #2: This manuscript presents results of applying a length-based model to estimate secondary production by Antarctic krill. The authors find that the model produces different estimates of production in different regions, and different years, and also produces different results when parameters are varied within plausible bounds. The authors discuss relevance to fisheries management.

It was unclear to me how the model worked, which made it difficult to evaluate the results.

I was also concerned by the assertion that length is a proxy for growth.

It is great to see fisheries data used, but I felt the manuscript was missing some mention of the limitations and potential biases in this data.

The title doesn't appear to reflect the contents, as most of the regions studied showed no evidence of temporal trends which could be associated with environmental change.

I found the manuscript difficult to read. The authors have clearly put a lot of effort into using elegant scientific language, but I found the unnecessary verbiage obscured the main messages of the manuscript. The manuscript could be much improved by simplifying the language and focusing on clearly communicating the approach and results.

In places, the language is also not exactly standard English (e.g. "Along the Southern Ocean" [the southern ocean is roughly a circle, one doesn't typically refer to going along circles], "Krill is the largest marine biomass on Earth" [krill may have or represent the largest biomass, but one doesn't typically use "is" in this context]).

The manuscript also contains many acronyms - readability could be improved by using these more judiciously, or if they are necessary, perhaps including a table or glossary.

The figures were also rather blurry in the proofs; perhaps this is an issue specific to the manuscript proofs, but it might be worth double checking, and considering if the plots could be output in a file format which better preserves resolution (e.g. SVG).

**Do you want your identity to be public for this peer review?** For information about this choice, including consent withdrawal, please see our Privacy Policy

Reviewer #1: No

Reviewer #2: No

---

## [Author Response · Author response to Decision Letter 2]

15 Dec 2025

Comments to the Author

1. If the authors have adequately addressed your comments raised in a previous round of review and you feel that this manuscript is now acceptable for publication, you may indicate that here to bypass the “Comments to the Author” section, enter your conflict of interest statement in the “Confidential to Editor” section, and submit your "Accept" recommendation.

Reviewer #1: All comments have been addressed

Reviewer #2: (No Response)

2. Is the manuscript technically sound, and do the data support the conclusions?

Reviewer #1: Yes

Reviewer #2: Partly

Reply: We appreciate the reviewer’s comment. The methodological details, including data templates, code, and all procedures used to ensure analytical rigor, are fully clarified in Supporting Information 1, where we provide complete documentation of the workflow and datasets employed.

3. Has the statistical analysis been performed appropriately and rigorously?

Reviewer #1: Yes

Reviewer #2: I Don't Know

Reply: We thank the reviewer for raising this point. The statistical formulas underlying the model are fully presented in Supporting Information 1, and all code used to implement the analyses is embedded within the manuscript to ensure full transparency and reproducibility.

4. Have the authors made all data underlying the findings in their manuscript fully available?

Reviewer #1: Yes

Reviewer #2: No

Reply: Thank you for your comment. As noted in the Data Availability Statement, he original raw data underlying this study are managed by the Commission for the Conservation of Antarctic Marine Living Resources (CCAMLR) and are subject to legal and institutional restrictions under the CCAMLR Rules for Access and Use of Data. As a result, the authors do not have permission to publicly share the original raw datasets.

However, the minimal processed and anonymized datasets required to replicate the analyses and results presented in this study are publicly available in a stable repository. These processed data, together with the complete analytical workflow, have been deposited in Zenodo and can be accessed via the following DOI: https://doi.org/10.5281/zenodo.17936869. The processed datasets archived in Zenodo consist of aggregated and atomized length–frequency data derived from the original CCAMLR datasets exclusively for the purpose of reproducibility and do not substitute the original raw data holdings.

Also, and to fully comply with PLOS data requirements while respecting these restrictions, we have made all processed data and the complete code needed to replicate the analysis are provided in the Code Repository section of Supporting Information 1, which offers a step-by-step guide for reproducing the results. In response to the reviewer’s observation, we have updated the README file in the code repository (https://github.com/MauroMardones/LBSPR_Krill linked to the Zenodo archive) to clearly explain the structure of the repository, the purpose of each script and dataset, and the workflow required to run the analysis.

5. Is the manuscript presented in an intelligible fashion and written in standard English?

Reviewer #1: Yes

Reviewer #2: No

Reply: The manuscript has undergone a thorough language revision to ensure clarity and fluency. All sections were revised to ensure that the text is clear, concise, and written in standard academic English, without introducing any structural changes to the original content. Although the reviewers did not specify which particular passages required language improvement, we have carefully revised the entire manuscript to address any potential issues. All modifications have been incorporated in the revised version using Track Changes to facilitate the review of the language improvements and ensure full transparency in the revision process.

6. Review Comments to the Author

Reviewer #1: All my comments have been addressed well. From my side, this manuscript was ready for publication.

Reviewer #2: This manuscript presents results of applying a length-based model to estimate secondary production by Antarctic krill. The authors find that the model produces different estimates of production in different regions, and different years, and also produces different results when parameters are varied within plausible bounds. The authors discuss relevance to fisheries management.

Reviewer 2 Observations

Obs1: It was unclear to me how the model worked, which made it difficult to evaluate the results.

Reply: We appreciate the reviewer’s observation and understand that the model description may not have been sufficiently clear in the original version. To improve the transparency and understanding of how the LBSPR model works and how was implemented for Antarctic krill, we have revised the Methods section and expanded the supplementary material.

In the revised version, we have added a set of equations in Supporting Information 1 that describe the full computational workflow of the SPR estimation process, from model inputs to intermediate calculations and final outputs. These equations outline the structure of the model and the sequence of operations in R codes used to obtain the estimates presented in the manuscript.

Obs 2: I was also concerned by the assertion that length is a proxy for growth.

Reply: We thank the reviewer for this comment. In fisheries science, using length as a proxy for growth is not only standard practice but foundational to the most widely applied growth and life-history models. Generally, growth can be defined as the change in length or weight over the life of an individual, and it directly influences survival, sexual maturity, reproductive success, movement, and migration (Peters, 1986). In this sense, the most commonly used model to describe somatic growth through time, von Bertalanffy Growth Function (VBGF), is explicitly length-based. This framework relies on observed lengths to estimate growth trajectories and key biological parameters such as the growth coefficient (k) and the asymptotic length (L∞). Thus, length is intrinsically embedded in the way growth is modeled, interpreted, and applied in fishery science. To strengthen this point, we have expanded the manuscript text (line 231) supporting the use of length as a proxy for growth in population assessments, including those related to the VBGF and length-based methods:

• Beverton, R. J. H., & Holt, S. J. (1957). On the Dynamics of Exploited Fish Populations. Fishery Investigations Series II, Vol. 19.

• Pauly, D. (1980). On the interrelationships between natural mortality, growth parameters, and environmental temperature in 175 fish stocks. ICES Journal of Marine Science, 39(2), 175–192.

• Sparre, P., & Venema, S. C. (1998). Introduction to Tropical Fish Stock Assessment. Part 1: Manual. FAO Fisheries Technical Paper 306/1.

• Peters, R. H. (1986). The Ecological Implications of Body Size. Cambridge University Press. (This reference have been added to the revised manuscript)

• Kell LT, Minto C, Gerritsen HD. Evaluation of the skill of length-based indicators to identify stock status and trends. ICES Journal of Marine Science. 2022;79: 1202–1216. doi:10.1093/icesjms/fsac043

Obs 3: It is great to see fisheries data used, but I felt the manuscript was missing some mention of the limitations and potential biases in this data.

Reply: We thank the reviewer for this helpful comment. In response, we have added a dedicated section in the Discussion called Limitations and caveats of length-based assessment methods addressing the limitations and potential uncertanty and biases associated with the fisheries data used in this study. This new text outlines key sources of uncertainty such as data scarse, representativeness of length data, and life history parameters. We have also included a discussion of the limitations of the LBSPR model itself, particularly regarding how estimation performance can vary depending on the stock’s status. Specifically, we now describe how SPR estimates may become biased when the population is either heavily depleted or lightly exploited, and how these conditions can influence the reliability of the derived biological indicators. We believe these additions aim to provide a more balanced interpretation of the results and to clarify the contexts in which the model performs robustly and where caution is needed.

Obs 4: The title doesn't appear to reflect the contents, as most of the regions studied showed no evidence of temporal trends which could be associated with environmental change.

Reply: Thank you for this comment. We agree that the observed temporal trends were not consistent across all regions, and in several cases no clear patterns emerged that could be directly linked to environmental change. However, the objective of the manuscript is not to demonstrate uniform temporal trends, but rather to highlight the high spatial variability in intrinsic productivity estimates and to discuss how such variability can occur even under a rapidly changing ocean.

The title, “Disparate estimates of intrinsic productivity for Antarctic krill across small spatial scales under a rapidly changing ocean,” emphasizes this central message, that intrinsic productivity varies markedly across neighbouring regions, and that this heterogeneity is highly relevant in the context of environmental change, even if not all regions show directional temporal trends. To improve clarity, we have adjusted the Introduction and Discussion to explicitly state that the primary focus of the study is spatial contrasts in intrinsic productivity rather than uniform temporal patterns.

Obs 5: I found the manuscript difficult to read. The authors have clearly put a lot of effort into using elegant scientific language, but I found the unnecessary verbiage obscured the main messages of the manuscript. The manuscript could be much improved by simplifying the language and focusing on clearly communicating the approach and results.

Reply: Thank you for this comment. We appreciate the concern regarding readability. In response, we conducted a detailed review of the entire manuscript, focusing on simplifying the language, reducing overly elaborate phrasing, and improving the clarity and flow of the text without compromising scientific accuracy. Because the reviewer did not specify which sections were problematic, we carried out a comprehensive, line-by-line assessment to identify and refine any wording that could obscure the main messages.

We also acknowledge that modelling-focused manuscripts can naturally be dense and demanding to read due to the technical nature of the methods and results. Even so, we made a concerted effort to improve clarity wherever possible, and we believe the manuscript is now more fluid and easier to follow. These improvements are highlighted in the tracked-changes file and also incorporated into the final revised manuscript.

Obs 6: In places, the language is also not exactly standard English (e.g. "Along the Southern Ocean" [the southern ocean is roughly a circle, one doesn't typically refer to going along circles], "Krill is the largest marine biomass on Earth" [krill may have or represent the largest biomass, but one doesn't typically use "is" in this context]).

Reply: Thank you for the observation. The expressions noted have been corrected in the revised manuscript, along with similar phrasing issues identified during our review (Lines 48 and 58)

Obs 7: The manuscript also contains many acronyms - readability could be improved by using these more judiciously, or if they are necessary, perhaps including a table or glossary.

Reply: We appreciate the reviewer’s observation. In response, we carefully reviewed the use of acronyms throughout the manuscript to ensure they are employed only where necessary. Additionally, Supporting Information 1 now includes a glossary of technical terms that defines the main parameters, variables, fishery concepts, and acronyms used in the analysis. We believe these improvements enhance the clarity of the document and strengthen the transparency and interpretability of the results.

Obs 8: The figures were also rather blurry in the proofs; perhaps this is an issue specific to the manuscript proofs, but it might be worth double checking, and considering if the plots could be output in a file format which better preserves resolution (e.g. SVG).

Reply: Thank you for your observation. The figures were included in the format required by the journal (.tiff). When downloaded directly from the PDF submmitted, they can be viewed at a higher resolution, which should address the issue you observed in the proofs.

7. PLOS authors have the option to publish the peer review history of their article (what does this mean?). If published, this will include your full peer review and any attached files.

Do you want your identity to be public for this peer review? For information about this choice, including consent withdrawal, please see our Privacy Policy.

Reviewer #1: No

Reviewer #2: No

---

## [Editor Report · Decision Letter 2]

8 Jan 2026

Disparate estimates of intrinsic productivity for Antarctic krill across small spatial scales, under a rapidly changing ocean.

PONE-D-25-16002R2

Dear Dr. MARDONES,

We’re pleased to inform you that your manuscript has been judged scientifically suitable for publication and will be formally accepted for publication once it meets all outstanding technical requirements.

Kind regards,

Clara F. Rodrigues

Academic Editor

PLOS One

Additional Editor Comments (optional):

The authors have systematically addressed the concerns raised during the review process, particularly regarding the transparency of the LBSPR model and the computational workflow. The inclusion of the supplementary equations and the glossary of technical terms significantly enhances the reproducibility and accessibility of the work. Furthermore, the newly added 'Limitations and Caveats' section provides a necessary and balanced perspective on the complexities of using length-based assessments for Antarctic krill. I appreciate the effort taken to refine the language and simplify the narrative, which has resulted in a much clearer communication of your findings regarding the spatial heterogeneity of intrinsic productivity. We believe this study makes a valuable contribution to the field of fisheries science in the Southern Ocean
---

## [Editor Report · Acceptance letter]

PONE-D-25-16002R2

PLOS One

Dear Dr. Mardones,

I'm pleased to inform you that your manuscript has been deemed suitable for publication in PLOS One. Congratulations! Your manuscript is now being handed over to our production team.

Kind regards,

on behalf of

Dr. Clara F. Rodrigues

Academic Editor

PLOS One